# What Motivates the Vaccination Rift Effect? Psycho-Linguistic Features of Responses to Calls to Get Vaccinated Differ by Source and Recipient Vaccination Status

**DOI:** 10.3390/vaccines11030503

**Published:** 2023-02-21

**Authors:** J. Lukas Thürmer, Sean M. McCrea

**Affiliations:** 1Department of Psychology, Paris-Lodron University Salzburg, 5020 Salzburg, Austria; 2Department of Psychology, University of Wyoming, Laramie, WY 82071, USA

**Keywords:** COVID vaccine hesitancy, experiment, societal divide, psycho-linguistic analyses, vaccination rift, Austria

## Abstract

Although vaccination provides substantial protection against COVID, many people reject the vaccine despite the opportunity to receive it. Recent research on potential causes of such vaccine hesitancy showed that those unvaccinated rejected calls to get vaccinated when they stemmed from a vaccinated source (i.e., a vaccination rift). To mend this vaccination rift, it is key to understand the underlying motivations and psychological processes. To this end, we used the voluntary free-text responses comprised of 49,259 words from the original Austrian large-scale data-set (*N* = 1170) to conduct in-depth psycho-linguistic analyses. These findings indicate that vaccinated message sources elicited longer responses using more words per sentence and simpler language writing more about things rather than themselves or addressing others directly. Contrary to common assumptions, expressed emotions or indicators of cognitive processing did not differ between message source conditions, but vaccinated sources led to more achievement-related expressions. Participant vaccination did not moderate the observed effects but had differential main effects on psycho-linguistic response parameters. We conclude that public vaccination campaigns need to take the vaccination status of the message source and other societal rifts into account to bolster recipients’ achievement.

## 1. Introduction

Evidence indicates that COVID-19 vaccines safely and effectively protect individuals from severe illness [1]. Nevertheless, many people in Western countries forego opportunities to receive the vaccine [2,3,4], partially because immunization program communications inadequately address such vaccine hesitancy [5]. Recent research indicates that group-level processes can help understand and potentially mend vaccine hesitancy. According to this view, vaccination status serves as a group membership and the resulting group boundaries cause rifts when communicating calls to get vaccinated between those vaccinated and those unvaccinated (i.e., a *vaccination rift effect* [6]). The vaccination rift was demonstrated in self-reports and behavior, but this past research is mute to the cognitions, emotions, and motives underlying this effect. We, therefore, use psycho-linguistic analyses to probe the processes underlying the vaccination rift.

Researchers’ warned early on during the COVID-19 pandemic that vaccine uptake may be insufficient unless informational campaigns target populations at risk for vaccine hesitancy [7]; these warnings, unfortunately, proved true [2]. The key determinant of decisions to get vaccinated is trust in the responsible health authorities [8,9,10], and confidence in vaccines and the system that delivers them [11]. Common vaccine communication strategies, such as using expert communicators, address the interpersonal, individual, and organizational levels to build trust [2]. However, as divisions in societies increase, trust decreases [12]. The resulting group lines thus represent mounting barriers to vaccination that go beyond the interpersonal, individual, and organizational levels (i.e., are at the group level). In line with this view, adherence to COVID-related health measures was substantially lower among Republicans than Democrats in the United States [13], leading to greater COVID-related mortality in Republican-leaning areas [14]. These relations between conservativism and vaccine hesitancy were also observed in Europe [15], pointing to a generalizable group-level phenomenon. Correlational research thus indicates that vaccination decisions are socially determined [3], with mounting group barriers representing a major impediment.

Recent experimental evidence indicates that these group barriers hinder critical communication about vaccination [16,17]. Group members typically tolerate critical comments from fellow (in)group members but reject the same criticism by a member of another (out)group [18]. Such intergroup criticism represents a violation of established conversational norms, leading perceivers to lose trust in the critic’s motives [19,20]. This lack of trust is so severe that perceivers will even invest their own money to punish intergroup criticism [20,21]. Critical communication across group lines thus results in a lack of trust in the message and the messenger [18,20,21], even among those who are not the target of the criticism (i.e., bystanders) [20,21,22]. This reasoning applies to vaccination decisions. Two sets of studies demonstrated that unvaccinated participants met calls to get vaccinated with suspicion when the messenger was an apparent outgroup member (i.e., expert from a state with different political preferences or a government official), in comparison to the same message from an apparent ingroup member (i.e., expert from a state with the same political preferences or an expert) [16,17]. Apparently, calls to get vaccinated across group lines are less effective.

Going beyond characteristics of certain groups, recent experimental research on the vaccination rift effect [6] argued that societal discourse about vaccination creates a group boundary between those vaccinated and those unvaccinated. Vaccination status itself thus becomes a social group membership, and this group boundary should impede critical communication such as calls to get vaccinated. This vaccination rift effect was recently demonstrated in a large (*N* = 1170), age-representative sample in Austria [6]. Participants rejected calls to get vaccinated by a vaccinated source as unconstructive and threatening, as compared to the same message from a so-far unvaccinated source. Participants also ascribed worse personality characteristics to vaccinated (vs. unvaccinated) commenters. The vaccination rift was substantially stronger among the critical group of unvaccinated participants than among those fully vaccinated. Indirect rift effects of message source on behavioral measures of vaccination planning and counterarguing via message motive ratings were observed. These measures are proximal predictors of behavior, indicating that the vaccination rift may impede vaccine uptake.

Moreover, Austria followed a high standard of vaccine availability and communication at the time of study. Structural barriers to vaccination (e.g., poor public health funding, insufficient communication or incentives, or low vaccine availability [3,23]) were very low and, therefore, cannot account for these findings. Austria has universal healthcare and vaccination was free to all citizens, with numerous incentives. Vaccination was offered at attractive sites (e.g., castles or airport hangars) and at varying locations close to peoples’ homes. Raffles as well as a vaccine mandate provided additional incentives to get vaccinated. Nevertheless, vaccination rates were relatively low, indicating that vaccine communication could have been improved [24]. In sum, Austria provided a good test setting.

However, the reported analyses only relied on pre-defined scales and behavioral items. It is, therefore, possible that results so far missed the influence of message source on other psychological processes. This leaves important questions unanswered, such as: What are participants trying to achieve with their responses? To whom are they addressed? Which cognitions, motivations and emotions do they convey? The words people use are indicators of how they see the world [25]. In line with this view, linguistic analyses can identify ongoing psychological processes such as personality [26], relations between groups [27], and processes within groups [28]. Linguistic analyses have also proven useful in investigating a range of phenomena, including group interaction processes [29], communication [30], and societal divides [31]. Analyzing the words people use rather than their responses to a Likert-type scale has the advantage that participants can give free responses. Linguistic analyses, therefore, provide the opportunity to explore psychological processes beyond existing theory. We, accordingly, explored psycho-linguistic processes underlying the vaccination rift effect in the content of participants’ free-text responses using automated quantitative linguistic analyses [25]. We, moreover, analyzed the influence of recipients’ own vaccination status on these processes, as well as a potential effect of matching the vaccination status of the source with the participant. We, thereby, seek to contribute to understanding vaccine hesitancy, why vaccination has become such a divisive topic, and how to communicate to mend the resulting societal divides.

Although no research to date has provided elaborate psycho-linguistic process analyses of vaccination rifts, past research on group criticism gives some indication regarding potential candidate processes. Past research has observed that participants invest more of their own time [32] and money [20,21,33] to refute outgroup criticism. This would suggest that the vaccination rift should lead to longer and more elaborate written responses (i.e., more words per response as well as longer sentences). A related question concerns who responds to vaccination rifts addresses. Responses to group criticism have been categorized as directed at the commenter, bolstering one’s own group, or derogating other groups [32]. Evidence in these studies was strongest for personal response towards the commenter. In the context of the vaccination rift, it is also possible that responses address the common public, as vaccination is an issue of societal importance. The psycho-linguistic approach allows quantifying these references by analyzing the frequency of pronouns used. In line with this view, pronoun use can signify one’s identity and has been associated with intergroup and intragroup processes [25,34,35,36].

Beyond the focus and quantity and syntax of communication, specific words can give an indication of different psychological processes. In the group criticism literature, one assumption is that responses to group criticism “are not guided by cold, rational examinations of the evidence” but “emotional, ‘hot’ processes, infused with threat, uncertainty and doubt” (p. 276, [18]). This suggests that vaccination rifts may reduce cognitive processes but increase emotional responding and “lashing out” [37]. Tentative evidence for this assertion comes from research on group criticism and costly punishment that indicated that the main motivation for responding to group criticism is retribution and not changing the commenter’s future behavior [21]. Alternatively, vaccination rifts may change the focus from making an informed decision to winning an argument. Past research on responses to social threats are in line with this view [38]. We used detailed psycho-linguistic analyses to test if and how the vaccination rift influences communication patterns and motivations.

We, thereby, aim to make three key contributions: First, the experimental study design allows identification of causal effects of message source vaccination status on psychological responses to calls to get vaccinated. Second, our large, age-representative sample of vaccinated and unvaccinated Austrian participants provides insights into the psychological processes of these societal groups. Importantly, the standardized study setting makes it unlikely that common biases, such as prejudice towards those unvaccinated [39], impact our findings. Finally, participants can respond to the free-text items in whichever way they chose. Our quantitative analysis of their responses, therefore, allows identification of processes underlying vaccine hesitancy that current theory does not cover. Jointly, these findings will identify new directions for theory on vaccine hesitancy and help design effective vaccine messages.

## 2. Materials and Methods

We report secondary analyses of the vaccination rift dataset [6]. The dataset was collected between 29 November 2021 and 1 December 2022 through news items in the national newspaper “Salzburger Nachrichten” on 29 November 2021 (online and print) and the public television station “ORF Salzburg” during the evening news “Salzburg heute” and on their website on 26 December 2021. As reported in the original publication, all ethical standards, including obtaining informed consent, as put forth by the Declaration of Helsinki and the American Psychological Association, were followed. The University of Salzburg internal review board approved this study (IRB number GZ 10/2020). All materials, data (except raw free-text responses that were removed due to privacy concerns), and analyses are available at https://osf.io/c8tn3/ (access date: 20 February 2023). 

We analyze a sample of *N* = 1170 (501 male, 663 female, 4 non-binary, 4 diverse; mean (*M*)_age_ = 49.09, standard deviation (*SD*) = 13.32, range [15; 83]; 1115 Austrian nationality; unvaccinated *N* = 370; recovered *N* = 166; single-dose vaccinated *N* = 15; double-dose vaccinated *N* = 153; fully vaccinated/triple-dose *N* = 466). One participant who withdrew consent after the study, 47 participants who failed one or more attention checks and 9 who reported not currently living in Austria (3 of whom also failed manipulation checks) were not included in this analysis sample. As reported [6], the sample age (*M* = 49.09, *SD* = 13.32, range [15; 83]) was representative of the adult population average (*M* = 49.74), *t*(1169) = −1.68, *p* = 0.094, *d* = −0.05, but oversampled female respondents (57% vs. 52% in the population, *z* = 3.43, *p* < 0.001) and undersampled male respondents (43% vs. 48% in the population, *z* = −3.43, *p* < 0.001). Fully vaccinated participants were well-represented (40% vs. 40% in the population as of 26 December 2021, *z* < 0.01, *p* > 0.999), but partly vaccinated or recovered participants were underrepresented (29% vs. 34% in the population, *z* = −3.63, *p* < 0.001), in exchange for an over-representation of unvaccinated participants (32% vs. 26% in the population, *z* = 4.65, *p* < 0.001). The experiment was conducted in the survey software formR [40], where participants were randomly assigned to an unvaccinated source or a vaccinated source condition.

We briefly summarize the methods of the original study; please refer to the vaccination rift publication [6] for further details. Participants read critical comments calling for unvaccinated people to get the COVID vaccine (see Appendix B). This comment was attributed to a former participant who was described as either being unvaccinated but intending to get vaccinated (unvaccinated source condition) or being fully vaccinated (vaccinated source condition). After reading each comment, participants completed manipulation checks on the source vaccination status, and rated message motive, message threat, and commenter characteristics. Participants then indicated if they wanted to receive more information on vaccination sites (yes/no). Key to the present analysis, participants then had the opportunity to respond to the free-text question: “What is your personal opinion about the Corona vaccination?” Participants could write as much or as little as they chose. Finally, participants provided demographic information (nationality, Austrian county of residence, vaccination status, age, and gender) and were debriefed. All participants then received official information on the COVID vaccine, as recommended by the ethics committee.

For the current analysis, we obtained the original dataset and analyzed the responses to the free-text question on participants’ opinion about the coronavirus vaccination. In line with recent recommendations [29], a research assistant blind to the studies’ hypotheses spell-checked and proof-read all comments. Obvious typographic errors were corrected but no other alterations were made to the texts. Specifically, all slang, swear words, and colloquial expressions were maintained. We then entered the free-text response into Linguistic Word Count and Inquiry (LIWC) 2015 [41], the industry-standard tool to conduct psycho-linguistic analyses. We used the 2015 German dictionary available for LIWC that has been validated and has measurement properties equivalent to the English version [42,43]. We used R [44] in R-Studio [45] including the packages psych [46], readxl, stringr, and dplyr to analyze our data.

## 3. Results

All linguistic measures were highly skewed, and we accordingly used a non-parametric Mann–Whitney–Wilcoxon tests for independent samples, applying continuity corrections. We, moreover, sought to find a balance between statistical power to observe (typically small) effects on linguistic markers and to control for alpha-error inflation due to multiple tests. We, therefore, decided to apply group-wise alpha corrections by dividing the threshold of *p* = 0.05 by the number of tests per category, yielding an effective alpha-level between 0.05 and 0.001. Moreover, we omit significant exploratory effects that were extremely small (i.e., medians of 0 across conditions). We indicate in our results section where tests were not significant due to the alpha-adjustment and report the full results in Appendix A. Non-parametric Spearman correlations between LIWC summary measures, source vaccination status and participant vaccination status are depicted in Figure 1; means and medians in Table 1.

### 3.1. Vaccination Rift

The starting point for our linguistic analyses was the observation that vaccinated sources’ critical calls to get vaccinated elicited longer responses than identical comments from an unvaccinated source [6].

We first sought to explore the potential rift in the complexity of participants’ responses. Compared to those who received a comment from the unvaccinated source, participants in the vaccinated source condition communicated more elaborately, that is, used more words per sentence (median (*Mdn*)_unvaccinated source_ = 10.71 vs. *Mdn*_vaccinated source_ = 11.34), *W* = 156,069, *mu* = −1.00, 95%CI [−2.00; −0.17], *p* = 0.009, and more simply, that is used fewer words with six letters or more (*Mdn*_unvaccinated source_ = 32.42 vs. *Mdn*_vaccinated source_ = 31.80), *W* = 184,382, *mu* = 1.62, 95%CI [0.03; 3.33], *p* = 0.022, and more words that were included in the LIWC2015 German dictionary (*Mdn*_unvaccinated source_ = 84.19 vs. *Mdn*_vaccinated source_ = 85.71), *W* = 156,998, *mu* = −1.25, 95%CI [−2.50; <−0.01], *p* = 0.014. These findings are in line with the view that the vaccination rift causes counter-arguing. Moreover, participants may have increased their effort to make themselves understood when communicating across group lines. The four broad summary scores provided by LIWC (Analytic, Clout, Authentic, and Tone) did not differ significantly between conditions, *W*s < 167,255, *p*s > 0.500. However, participants in the vaccinated source condition used more personal pronouns (*Mdn*_unvaccinated source_ = 3.85 vs. *Mdn*_vaccinated source_ = 4.93), *W* = 157,896, *mu* < −0.01, 95%CI [−0.18; <−0.01], *p* = 0.019. Follow-up analyses indicated that this vaccination rift effect especially emerged for impersonal pronouns (*Mdn*_unvaccinated source_ = 3.85 vs. *Mdn*_vaccinated source_ = 5.26), *W* = 148,746, *mu* = −0.14, 95%CI [−1.26; <−0.01], *p* < 0.001. Effects of 3. Person plural pronouns (addressing others) were significant but small; effects for 1. Person pronouns addressing the writer themselves (“I”) or their group (“we”) were non-significant, as were 2. Person pronouns addressing others directly (“you”). Apparently, the vaccination rift leads to communicating about things rather than with each other. Interestingly, no effects of “we” and “you” emerged. These pronouns are typical indicators of processes within and between groups such as cooperation and conflict [34,35]. The absence of rift effects on these linguistic markers indicates that a strong identification with the group of the unvaccinated may not be a prerequisite for vaccine hesitancy. This conclusion would be in line with recent research indicating that group identification does not influence the rejection of outgroup criticism [20,21].

We next explored the emotional, cognitive, and motivational processes underlying the vaccination rift effect in more detail. Social conflict is commonly assumed to hinge on emotional processes with reduced cognitive processing. Contrary to this popular assumption, we did not observe a vaccination rift effect in the emotional content of the free-text responses (*Mdn*_unvaccinated source_ = 4.74 vs. *Mdn*_vaccinated source_ = 5.56), *W* = 160,304, *mu* < −0.01, 95%CI [−0.54; <0.01], *p* = 0.058. Exploratory follow-up analyses on positive emotions, negative emotions, as well as specific negative emotions (anger, anxiety, and sadness) all did not indicate a vaccination rift, *W*s < 162,994, *p*s > 0.140. Finally, the vaccination rift also did not emerge on indicators of cognitive processes (*Mdn*_unvaccinated source_ = 25.00 vs. *Mdn*_vaccinated source_ = 24.90), *W* = 178,236, *mu* = 0.74, 95%CI [−0.33; 2.30], *p* = 0.217. These findings indicate that vaccination rifts may rely less on increased emotional and reduced cognitive processes that previously assumed. It is, thus, not the case that the rejection of calls to get vaccinated stems from “all feeling and no thinking”.

Taking an alternative perspective, outgroup criticism may violate participants’ basic needs. Indeed, a vaccination rift effect on motivational indicators emerged, such that comments from the unvaccinated source led to using fewer drive-related words (*Mdn*_unvaccinated source_ = 7.69) as compared to participants who received a comment from the vaccinated source (*Mdn*_vaccinated source_ = 9.01), *W* = 159,022, *mu* = −0.27, 95%CI [−1.41; <−0.01], *p* = 0.035. Follow-up analyses indicated that this effect emerged for achievement-related words (*Mdn*_unvaccinated source_ = 0, *Mdn*_vaccinated source_ = 1.73), *W* = 156,967, *mu* < −0.01, 95%CI [<−0.01; <−0.01], *p* = 0.009. No effects emerged on words related to affiliation, power, risk, or reward, all *W*s < 163,328, all *p*s > 0.150. Apparently, the vaccination rift effect hinges on perceptions of competence and achievement more than affiliation or shifting risk-perceptions. Calls to get vaccinated from vaccinated sources thus cause rifts because they lead to debates about competence, for instance whether the vaccine and those promoting it are efficacious. Once these questions about the source arise, it is difficult to process new information in an unbiased manner.

Finally, exploratory analyses on perception processes, biological processes, time focus, relativity, and informal language did not show any effects, all *W*s < 160,679, all *p*s > 0.059. Responses to the vaccinated source referred more to social processes (*Mdn*_unvaccinated source_ = 9.09, *Mdn*_vaccinated source_ = 10.09), *W* = 153,742, *mu* = −0.74, 95%CI [−2.11; <−0.01], *p* = 0.002, but no effects on the subcategories family, friends, female, and male emerged in follow-up analyses, all *W*s < 166,474, all *p*s > 0.110.

To summarize, a vaccination rift effect was evident in the linguistic markers, such that responses to vaccinated (vs. unvaccinated) sources:
Were more elaborate (longer and more words/sentence), but simpler (fewer words with six letters or more and more words in the dictionary)Addressed things more, but there was no indication for group processesWere not more emotional or less thought-through but included more drive-related words, especially those related to achievement, and more words related to general social processes.

### 3.2. Participant Vaccination Status

Our next goal was to test whether responses varied based on participants’ own vaccination status, collapsed across message source conditions. To this end, we clustered participants into two vaccination status groups: vaccinated (1, 2, or 3 doses, *n* = 634) and unvaccinated (recovered or unvaccinated, *n* = 536). For exploratory purposes, we also compared the key subgroups of fully vaccinated (3 doses, *n* = 466) and unvaccinated participants (*n* = 370). Results between the full sample and this subsample did not differ, unless noted otherwise.

We first sought to explore the length and complexity of participants’ responses. Compared to those who indicated that they were vaccinated, unvaccinated participants wrote more (*Mdn*_unvaccinated participant_ = 31.50 vs. *Mdn*_vaccinated participant_ = 15.00), *W* = 226,893, *mu* = 14.00, 95%CI [11.00; 17.00], *p* < 0.001, and used more words per sentence (*Mdn*_unvaccinated participant_ = 12.43 vs. *Mdn*_vaccinated participant_ = 10.00), *W* = 205,690, *mu* = 2.50, 95%CI [1.71; 3.17], *p* < 0.001. Unvaccinated participants used fewer words with six letters or more (*Mdn*_unvaccinated participant_ = 30.93 vs. *Mdn*_vaccinated participant_ = 33.33), *W* = 153,199, *mu* = −2.20, 95%CI [−3.70; −0.59], *p* < 0.001. No effect emerged on the use of words included in the dictionary (*Mdn*_unvaccinated participant_ = 84.48 vs. *Mdn*_vaccinated participant_ = 85.71), *W* = 160,938, *mu* = −0.69, 95%CI [−1.93; <0.01], *p* = 0.119. However, when excluding partly vaccinated and recovered participants, a small effect emerged. Unvaccinated participants used somewhat fewer dictionary words than those fully vaccinated (*Mdn*_unvaccinated participants only_ = 83.99 vs. *Mdn*_fully vaccinated participants only_ = 85.71), *W* = 78,999, *mu* = −1.36, 95%CI [−2.93; <−0.01], *p* = 0.037. Apparently, in comparison to vaccinated participants, those unvaccinated responded more to the call to get vaccinated and used somewhat simpler (shorter) words. These effects were thus quite similar to those of source vaccination status.

We next analyzed the four broad summary scores provided by LIWC. Unvaccinated participants used language that was less analytic (*Mdn*_unvaccinated participant_ = 37.27 vs. *Mdn*_vaccinated participant_ = 64.06), *W* = 146,281, *mu* = −2.41, 95%CI [−7.24; <−0.01], *p* < 0.001, less confident (*Mdn*_unvaccinated participant_ = 45.33 vs. *Mdn*_vaccinated participant_ = 61.77), *W* = 144,220, *mu* = −6.94, 95%CI [−10.57; −3.37], *p* < 0.001, and less positive in tone (*Mdn*_unvaccinated participant_ = 17.39 vs. *Mdn*_vaccinated participant_ = 17.39), *W* = 140,151, *mu* < −0.01, 95%CI [−0.61; <−0.01], *p* < 0.001. However, messages by unvaccinated participants used more authentic language (*Mdn*_unvaccinated participant_ = 41.77 vs. *Mdn*_vaccinated participant_ = 16.53), *W* = 186,943, *mu* < 0.01, 95%CI [<0.01; 3.29], *p* = 0.003. These observations comport well with the assumption that calls to get vaccinated represent a personal threat to those unvaccinated, independent of the source of the calls. Moreover, unvaccinated participants in the study apparently responded quite freely to our open-ended questions, as indicated by higher authenticity scores.

Unvaccinated participants used more personal pronouns (*Mdn*_unvaccinated participant_ = 4.79 vs. *Mdn*_vaccinated participant_ = 3.96), *W* = 185,189, *mu* < 0.01, 95%CI [<0.01; 0.65], *p* = 0.006. This effect again emerged most prominently for impersonal pronouns (*Mdn*_unvaccinated participant_ = 6.25 vs. *Mdn*_vaccinated participant_ = 1.98), *W* = 223,006, *mu* = 2.94, 95%CI [2.22; 3.57], *p* < 0.001. Effects of 3. Person plural pronouns (addressing others) were significant but small; effects for 1. Person pronouns addressing the writer themselves (“I”) or their group (“we”) were non-significant, as were 2. Person pronouns addressing others directly (“you”). Just as the observed message source effects, unvaccinated participants communicated about things rather than themselves or with each other. Interestingly, again, no group process effects of “we” and “you” emerged, indicating that the divide between those vaccinated and those unvaccinated may not hinge on strong group identification.

We next explored the emotional, cognitive, and motivational expressions in more detail. In line with the observations regarding emotional tone, the free-text responses by unvaccinated participants contained more negative emotion words (*Mdn*_unvaccinated participant_ = 1.68 vs. *Mdn*_vaccinated participant_ = 0), *W* = 205,981, *mu* < 0.01, 95%CI [<0.01; <0.01], *p* < 0.001, and fewer positive emotion words (*Mdn*_unvaccinated participant_ = 1.94 vs. *Mdn*_vaccinated participant_ = 2.27), *W* = 156,518, *mu* < −0.01, 95%CI [<−0.01; <−0.01], *p* < 0.001. Follow-up analyses observed small but significant differences on all three specific negative emotions (anger, anxiety, and sadness). Those unvaccinated thus feel strongly about the topic of vaccination.

Finally, no effects of participant vaccination status emerged on indicators of cognitive processes (*Mdn*_unvaccinated participant_ = 24.00 vs. *Mdn*_vaccinated participant_ = 24.20), *W* = 167,697, *mu* < 0.01, 95%CI [−1.67; 1.01], *p* = 0.701, or drive-related words (*Mdn*_unvaccinated participant_ = 8.27, *Mdn*_vaccinated participant_ = 8.33), *W* = 165,201, *mu* < −0.01, 95%CI [−0.73; <0.01], *p* = 0.410. Exploratory follow-up analyses indicated small effects (i.e., all *Mdn* = 0) on achievement and power but no effects on affiliation or reward. Overall, threatened needs seem less important for effects of vaccination status than for the vaccination rift effect. Indicators of cognitive processes again showed no effect.

Exploratory analyses indicated that unvaccinated participants referred more to biological processes, perception processes, and informal language, but these effects were small (i.e., all *Mdn* = 0). Unvaccinated participants used more words establishing the relation between different objects or people (*Mdn*_unvaccinated participant_ = 11.97, *Mdn*_vaccinated participant_ = 10.27), *W* = 184,706, *mu* = 0.42, 95%CI [<0.01; 1.83], *p* = 0.009, including words related to motion (*Mdn*_unvaccinated participant_ = 0, *Mdn*_vaccinated participant_ = 0), *W* = 192,272, *mu* < 0.01, 95%CI [<0.01; <0.01], *p* < 0.001, and space (*Mdn*_unvaccinated participant_ = 6.89, *Mdn*_vaccinated participant_ = 4.76), *W* = 193,104, *mu* = 0.50, 95%CI [<0.01; 1.67], *p* < 0.001. Effects in relation to time were inconsistent between the full and sub-sample. Unvaccinated participants used more words related to social settings (*Mdn*_unvaccinated participant_ = 10.64, *Mdn*_vaccinated participant_ = 8.84), *W* = 193,390, *mu* = 1.68, 95%CI [0.30; 3.02], *p* < 0.001, especially male words, although this effect was very small (i.e., all *Mdn* = 0). No effects emerged for female, family, and friends words, all *W*s < 173,528, all *p*s > 0.180. Apparently, vaccination status has an impact on the social world of those who are unvaccinated. This interpretation is in line with recent research on prejudice towards those unvaccinated [39]. We return to this point in the discussion.

To summarize, in comparison to vaccinated participants, unvaccinated participants responses to calls to get vaccinated:
Were more elaborate (longer and more words/sentence), but simpler (fewer words with six letters or more)Were less analytic, less confident, and less positive in tone, but more authenticAddressed things more, but showed no greater indicators of group processesShowed less positive and more negative emotionsReferred more to spatial as well as social relations.

### 3.3. Matching of Participant and Source Vaccination Status

A final possibility is that the congruence between participant and source vaccination status influences the psycho-linguistic processes in responses to calls to get vaccinated. We explored this possibility by coding whether participant and source status matched (unvaccinated source and unvaccinated participant, vaccinated source and vaccinated participant, *n* = 564) or not (unvaccinated source and vaccinated participant, vaccinated source and unvaccinated participant, *n* = 606). Note that this comparison is akin to an omnibus interaction test in parametric analyses (e.g., ANOVA).

We observed no consistent effects of the matching variable on any of the dependent measures, *W*s < 181,032, *p*s > 0.075, with the following two exceptions: Participants whose vaccination status matched the source vaccination status used more words that were in the LIWC dictionary (*Mdn*_match_ = 85.50, *Mdn*_mismatch_= 84.04), *W* = 183,475, *mu =* 1.07, 95%CI [<0.01; 2.34], *p* = 0.029, and more words related to anger (*Mdn*_match_ = 0, *Mdn*_mismatch_ = 0), *W* = 178,651, *mu* < 0.01, 95%CI [<0.01; <0.01], *p* = 0.018. When only including fully vaccinated and unvaccinated participants, small effects on the summary measure of analytic language (*Mdn*_match_ = 52.10, *Mdn*_mismatch_ = 55.60), *W* = 94,556, *mu* < 0.01, 95%CI [<−0.01; <0.01], *p* = 0.028, and on “we” personal pronouns emerged (*Mdn*_match_ = 0, *Mdn*_mismatch_ = 0), *W* = 81,712, *mu* < −0.01, 95%CI [<−0.01; <0.01], *p* = 0.028, and the observed differences on dictionary words and anger were no longer significant, *W*s < 90,787, *p*s > 0.050.

To summarize:
The matching of source and participant vaccination status had no consistent effects on the observed psycho-linguistic processes.Apparently, the vaccination rift elicits psycho-linguistic processes that are independent of those elicited by recipients’ own vaccination status.

## 4. Discussion

We provide in-depth linguistic analyses of the vaccination rift experiment, relying on 49,259 words in voluntary free-text responses from an age-representative sample of the Austrian population (*N* = 1170). Calls to get vaccinated from vaccinated sources caused rifts in psycho-linguistic processes, in comparison to the same calls from so-far unvaccinated sources. Experimental manipulation of the message source provides the gold-standard for causal inferences.

Participants’ vaccination status also had consistent effects that were different from those of message source. While message source effects focused on drives, especially achievement and social processes, participant vaccination status influenced a range of psycho-linguistic processes but not drive-related words. This pattern of results suggests that the message source, rather than participants’ vaccination status, elicits discussions about competence and efficacy of vaccines. Although the findings regarding participants vaccination status were correlational, robustness-checks between the full and a subsample increase our confidence in the observed effects.

Finally, we did not observe consistent effects of the matching between source and participant vaccination status. This may be surprising since the vaccination rift effect on self-report scales has been reported to be substantially larger among unvaccinated participants [6]. Psycho-linguistic analyses are arguably more unobtrusive than presenting self-report scales. Accordingly, one may argue that our current analyses tap into more subtle processes that are more ubiquitous. In line with this view, rejection of criticism has even been observed among bystanders who are not members of the criticized groups [20,21,22,47,48]. This indicates that choosing message sources for calls to get vaccinated carefully may even benefit those already vaccinated.

Interestingly, there was little evidence that source and participant vaccination status influenced discussions about group membership (i.e., “we” and “they”) or personal identity (“I”). Apparently, strong group memberships are no pre-requisite for rifts to emerge. One may consider this good news, since vaccination may not instill strong group boundaries; this also indicates, however, that vaccination rifts do not only emerge among highly identified fanatics but represent a common social process (cf. [49,50]). The vaccination rift effect could thus represent a much greater impediment to vaccination than previously assumed.

There are some limitations to our study that warrant discussion. First, it is important to note that the observed effect sizes were small. Effects of this size, however, are in line with past research using psycho-linguistic analyses [51]. Considering the volatility in free responses, it is remarkable how consistent the observed patterns were. Second, our dataset just contained participants from Austria. This setting was well-suited for the investigation (e.g., low structural barriers to vaccination, substantial civic liberties). The dataset is nevertheless mute to whether the observed processes occur in other cultures. Given the large variation of vaccine acceptance between different countries [52,53], studies in other cultural contexts are clearly needed. Third, we used an experimental design. While experiments allow drawing gold-standard conclusions about the causal direction of effects, they are mute to within-person changes. Similarly, the media channels advertised our study broadly, but it is possible that extreme vaccine critics may not have participated. Such a selection bias may have diminished the observed effects. Future longitudinal research with extreme vaccine critics is, therefore, needed to understand the temporal dynamics of responses to calls to get vaccinated. Finally, our study was conducted during a time when vaccination was at the center of public debate. For instance, a vaccine mandate had been issued in Austria. This attention may have increased vaccine messaging effects among those who are already vaccinated. More broadly, we explicitly referred to COVID-19 vaccination during an ongoing global pandemic. Future research should thus investigate if our results generalize to settings where COVID-19 vaccination is not at the center of public debate and to other vaccination decisions.

More broadly, our analysis provides key insights into the social determinants of vaccine hesitancy. Beyond low education and awareness or lacking opportunities [52], vaccination decisions are closely linked to one’s social system. In line with this view, cultural norms and peer vaccination decisions have been observed to influence vaccine hesitancy [54]. This observation is mirrored in our findings regarding social process. It is, therefore, key to take the social world into account when designing vaccine communication programs.

Trust and other general beliefs and attitudes towards vaccination are key for vaccine acceptance [55,56], but the vaccination rift diminishes trust [6]. The current analysis sheds light on the underlying processes: Participants’ counterarguing focused on achievement, highlighting the role of competence-related issues and questions. Unvaccinated participant moreover showed that the vaccine is a highly emotional topic for them that is approached with less confidence and less analytically. Considering these response patterns, it is hardly surprising that vaccine uptake is low in this population.

In extension, we speculate, that effects of participant vaccination status are a consequence of repeated rift experiences. For instance, recent research has reported extreme discriminatory attitudes against the unvaccinated [39]. Such discrimination may impede, rather than promote, vaccine uptake. Others have observed that expressed negativity can deter others from getting vaccinated [57]. We observed that those unvaccinated expressed substantial negative affect when writing about the COVID-19 vaccine, which may be indicative of these processes in free communication.

Our findings have direct implications for public health policy. Taking the vaccination status of the message source into consideration is easy to implement at scale. Ongoing campaigns could readily be adjusted, and the rift approach can be combined with existing effective approaches. For instance, using expert communicators has been shown to increase message effectiveness [53]. Unfortunately, it is all too common for expert communicators to highlight their early adoption of the vaccine and their status as being vaccinated. Our research indicates that it would be beneficial to also communicate doubts of recently-swayed sceptics. Our research thus indicates that relatively minor changes in vaccine communication may have a substantial impact on people’s trust and their responses. This finding highlights the need to convince rather than coerce those who are skeptical of public health interventions and to respect societal group boundaries.

## 5. Conclusions

Getting vaccinated saves countless lives but a substantial minority chooses to forego this opportunity. The current analysis indicates that the rifts along the lines of vaccination status emerged even in the words people use to write about the vaccine. This indicates that vaccine hesitancy is a substantial societal problem. However, our analysis also contributes to understanding these basic processes. We hope that applying this knowledge will promote constructive debates and mend the rifts that COVID-19 has torn.

## Figures and Tables

**Figure 1 vaccines-11-00503-f001:**
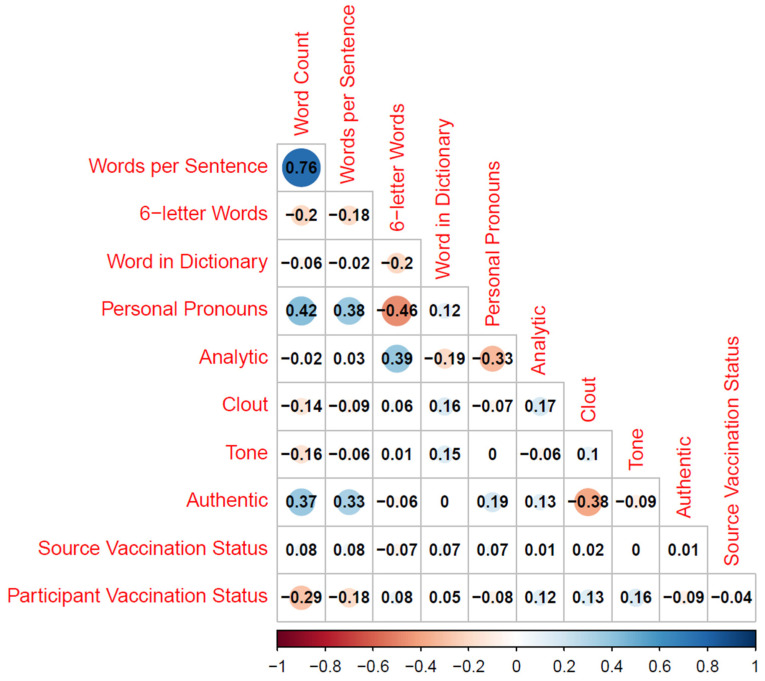
Spearman correlations between linguistic summary scores, source vaccination status and participant vaccination status. Note. Vaccination status coded as −1: unvaccinated; 1: vaccinated.

**Table 1 vaccines-11-00503-t001:** Means and medians by Source and Participant vaccination status ^1^.

	Unvaccinated/Recovered Participants				Vaccinated Participants			
	Source				Source			
	Unvaccinated		Vaccinated		Unvaccinated		Vaccinated	
	M (SD)	Mdn [Range]	M (SD)	Mdn [Range]	M (SD)	Mdn [Range]	M (SD)	Mdn [Range]
	Summary Scores and Grammar							
Word count	54.28 (69.60)	30 [0; 516]	53.61 (81.75)	33 [0; 1101]	24.14 (34.65)	14 [0; 294]	32.03 (64.60)	17 [0; 786]
Words per sentence	12.53 (6.88)	12.39 [0; 29]	13.08 (7.12)	12.43 [0; 52]	9.95 (7.50)	9 [0; 54]	11.39 (7.56)	10.94 [0; 43]
6-letter words	34.93 (19.69)	31.77 [0; 100]	30.72 (14.09)	30.35 [0; 100]	38.17 (21.93)	33.33 [0; 100]	34.81 (20.23)	32.84 [0; 100]
Words in dictionary	80.98 (18.58)	84.62 [0; 100]	83.33 (12.20)	84.34 [0; 100]	80.92 (18.07)	83.59 [0; 100]	85.07 (13.39)	86.67 [0; 100]
Analytic	46.47 (39.83)	41.25 [0; 99]	43.93 (39.43)	34.88 [0; 99]	52.23 (42.07)	57.23 [0; 99]	57.31 (41.22)	75.12 [0; 99]
Clout	42.89 (27.13)	42.96 [0; 99]	46.14 (28.90)	46.52 [0; 99]	51.49 (27.67)	61.81 [0; 99]	51.53 (28.92)	57.43 [0; 99]
Authentic	46.96 (40.14)	44.95 [0; 99]	43.84 (38.32)	39.33 [0; 99]	37.48 (41.02)	16.77 [0; 99]	38.67 (40.57)	16.23 [0; 99]
Tone	35.18 (36.90)	17.39 [0; 99]	34.48 (37.67)	17.39 [0; 99]	43.89 (40.76)	17.39 [0; 99]	47.18 (41.92)	17.39 [0; 99]
Personal pronouns	5.46 (5.82)	4.55 [0; 33.33]	6.30 (5.98)	5.12 [0; 33.33]	5.23 (6.89)	2.95 [0; 50]	5.75 (6.52)	4.65 [0; 36.36]
Impersonal pronouns	6.65 (6.84)	5.58 [0; 50]	8.15 (7.89)	6.85 [0; 50]	4.24 (8.33)	0.00 [0; 100]	5.13 (6.96)	2.82 [0; 50]
	Emotion and Cognition							
Positive emotion	3.21 (5.64)	2.06 [0; 50]	3.11 (5.00)	1.89 [0; 50]	7.67 (16.97)	0.79 [0; 100]	8.80 (18.79)	2.76 [0; 100]
Negative emotion	3.06 (7.60)	1.87 [0; 100]	3.11 (7.19)	1.44 [0; 100]	1.55 (3.11)	0.00 [0; 20]	1.76 (3.18)	0.00 [0; 25]
Cognitive processes	27.03 (16.09)	25.00 [0; 100]	25.48 (14.40)	23.17 [0; 100]	29.40 (23.11)	25.00 [0; 100]	28.62 (21.61)	24.07 [0; 100]
	General Processes							
Social	10.56 (8.94)	10.00 [0; 50]	12.31 (8.90)	11.11 [0; 50]	9.12 (9.70)	9.00 [0; 50]	10.74 (11.03)	9.58 [0; 60]
Perception	0.98 (1.89)	0 [0; 14.29]	0.87 (1.58)	0 [0; 12.50]	1.18 (6.12)	0 [0; 100]	2.03 (10.00)	0 [0; 100]
Biological	3.96 (5.35)	2.88 [0; 50]	4.09 (4.49)	3.45 [0; 33.33]	3.62 (6.20)	0 [0; 50]	4.13 (6.75)	1.94 [0; 50]
Relative	11.98 (10.73)	11.90 [0; 75]	11.89 (8.40)	12.04 [0; 66.67]	10.43 (10.63)	10.00 [0; 50]	11.23 (10.13)	10.35 [0; 50]
Informal	1.55 (8.87)	0 [0; 100]	1.62 (7.37)	0 [0; 100]	1.71 (8.27)	0 [0; 100]	2.82 (12.76)	0 [0; 100]
Drives/Motives	8.85 (8.77)	7.84 [0; 100]	9.23 (8.58)	8.33 [0; 100]	11.66 (16.72)	7.24 [0; 100]	11.85 (14.28)	9.09 [0; 100]

^1^ standard deviations are in parentheses, ranges are in brackets.

## Data Availability

Data, materials, and analyses are available at https://osf.io/c8tn3/.

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
