# Peer review of "What Motivates the Vaccination Rift Effect? Psycho-Linguistic Features of Responses to Calls to Get Vaccinated Differ by Source and Recipient Vaccination Status"

_vaccines, 2023, doi:10.3390/vaccines11030503_

Round 1

Reviewer 1 Report

Authors investigated motivation factors for the vaccination rift effect in a sample of Austrian citizens. The manuscript is interesting and well written adding valuable information on the field. I suggest some minor revisions to authors.

Lines 145-146

Please use the full date, e.g. 29 November 2021 instead of 11/29/2021

Lines 136-142

Could you please shorten the aim of your study avoiding details such as the adjustment for the alpha-level? Could you also make clear the gap that your study fill in the literature?

Lines 154-155

Please, for the first time use the full term, e.g. mean (M)=

Lines 154-155 And lines 187

Please do not repeat these results

Lines 196

Please avoid mixing methods with results.

Lines 197-205

Please remove this information in the methods section and then describe thoroughly the statistical analysis you performed, e.g. z test, t test, wilcoxon test, correlation coefficient.

Figure 1

Which correlation coefficient did you calculate?

Lines 214

Please avoid using references in the Results section

Lines 240-247, 365-367

Please do not mix results with discussion

Please add the Limitations of your study.

Please expand your conclusions describing the effect of your results on the public health policies.

Author Response

Responses to Reviewer 1 Comments.

Rev 1, Comment 1: Authors investigated motivation factors for the vaccination rift effect in a sample of Austrian citizens. The manuscript is interesting and well written adding valuable information on the field. I suggest some minor revisions to authors.

Author Response: Thank you for your thorough and positive evaluation of our paper. We respond to each of your comments below.

Rev 1, Comment 2: Lines 145-146: Please use the full date, e.g. 29 November 2021 instead of 11/29/2021

Author Response: We have changed the dates accordingly.

Rev 1, Comment 3: Lines 136-142: Could you please shorten the aim of your study avoiding details such as the adjustment for the alpha-level? Could you also make clear the gap that your study fill in the literature?

 Author Response: We have moved the information on language coding and on alpha adjustment to the method section (Lines 206-208), and we have added a paragraph on our study aims and impact. (Lines 136-146)

Rev 1, Comment 4: Lines 154-155: Please, for the first time use the full term, e.g. mean (M)=

Author Response: We now introduce mean, median, and standard deviation.

Rev 1, Comment 5: Lines 154-155 And lines 187: Please do not repeat these results

 Author Response: We have combined both paragraphs in the methods section.

Rev 1, Comment 6: Lines 196: Please avoid mixing methods with results.

  Author Response: We have removed the reference to LIWC.

Rev 1, Comment 7: Lines 197-205: Please remove this information in the methods section and then describe thoroughly the statistical analysis you performed, e.g. z test, t test, wilcoxon test, correlation coefficient.

   Author Response: We have updated this section with additional information on the tests performed.

Rev 1, Comment 8: Figure 1: Which correlation coefficient did you calculate?

  Author Response: We have updated the figure with non-parametric Spearman correlations and now specify this in the capitation.

Rev 1, Comment 9: Lines 214: Please avoid using references in the Results section

   Author Response: We would like to retain this reference to make clear that this result has been published elsewhere.

Rev 1, Comment 10: Lines 240-247, 365-367: Please do not mix results with discussion

   Author Response: We agree with your comment but sought to follow the Vaccines author instruction to “provide a concise and precise description of the experimental results, their interpretation, as well as the experimental conclusions that can be drawn.” (emphasis ours). We ask the editor to clarify.

Rev 1, Comment 11: Please add the Limitations of your study.

   Author Response: We now highlight several limitations of our study (Lines 439-459)

Rev 1, Comment 12: Please expand your conclusions describing the effect of your results on the public health policies.

   Author Response: We now highlight several implications of our study for public health policy (Lines 491-505). Thank you again for serving as a reviewer on our manuscript.

Reviewer 2 Report

Thank you for sharing this article with me. This is an important topic. I have several minor comments.

1. Please indicate in the title and abstract that this study was conducted in Austria.

2. I recommend to include some contextual information on COVID-19 health communication strategies in the context of Austria in Introduction so that you can talk about study findings within the policy context in Discussion.

3. Please discuss the limitation linked to selection bias. The survey is selecting respondents with some characteristics, potentially leading to biased estimates. Similarly, this is a cross-sectional survey, which comes with a long list of limitations. Please discuss these and provide some recommendations for future research. 

Author Response

Responses to Reviewer 2 Comments.

Rev 2, Comment 1: Thank you for sharing this article with me. This is an important topic. I have several minor comments.

Author Response: Thank you for serving as a reviewer on our manuscript. We respond to your comments below.

Rev 2, Comment 2: 1. Please indicate in the title and abstract that this study was conducted in Austria.

Author Response: We now indicate in the abstract that the sample is Austrian.

Rev 2, Comment 3: 2. I recommend to include some contextual information on COVID-19 health communication strategies in the context of Austria in Introduction so that you can talk about study findings within the policy context in Discussion.

Author Response: We now discuss the study setting with regard to COVID vaccination in more detail (Lines 83-92)

Rev 2, Comment 3: 3. Please discuss the limitation linked to selection bias. The survey is selecting respondents with some characteristics, potentially leading to biased estimates. Similarly, this is a cross-sectional survey, which comes with a long list of limitations. Please discuss these and provide some recommendations for future research. 

Author Response: We have expanded our discussion of potential limitations (Lines 439-459) and highlight the potential selection bias as well as the cross-sectional nature of our experiment. Thank you again for serving as a reviewer.